# Novel RNA viruses associated with avian haemosporidian parasites

**Jose Roberto Rodrigues** **, Scott W. Roy\*, Ravinder N. M. Sehgal\***

Department of Biology, San Francisco State University, San Francisco, CA, United States of America

\* sehgal@sfsu.edu (RNMS); scottwroy@gmail.com (SWR)

## Abstract

Avian haemosporidian parasites can cause malaria-like symptoms in their hosts and have been implicated in the demise of some bird species. The newly described Matryoshka RNA viruses (MaRNAV1 and MaRNAV2) infect haemosporidian parasites that in turn infect their vertebrate hosts. MaRNAV2 was the first RNA virus discovered associated with parasites of the genus *Leucocytozoon*. By analyzing metatranscriptomes from the NCBI SRA database with local sequence alignment tools, we detected two novel RNA viruses; we describe them as MaRNAV3 associated with *Leucocytozoon* and MaRNAV4 associated with *Parahaemoproteus*. MaRNAV3 had ~59% amino acid identity to the RNA-dependent RNA-polymerase (RdRp) of MaRNAV1 and ~63% amino acid identity to MaRNAV2. MaRNAV4 had ~44% amino acid identity to MaRNAV1 and ~47% amino acid identity to MaRNAV2. These findings lead us to hypothesize that MaRNAV_like viruses are widespread and tightly associated with the order Haemosporida since they have been described in human *Plasmodium vivax*, and now two genera of avian haemosporidians.

## Introduction

Avian haemosporidian parasites are found worldwide and over 1,500 different lineages have been described [1]. These parasites fall under the phylum Apicomplexa and include the genera *Plasmodium*, *Leucocytozoon*, and *Haemoproteus* [2]. Although all these parasites have different life cycles and vectors, they can cause malaria-like symptoms in the avian hosts. Species of avian haemosporidian parasites have been shown to exhibit variation in how they affect the host, with some cases being barely detectable and others resulting in severe anemia and death [3–5]. Naive avian communities have been found to be especially susceptible to haemosporidian infections, such as those in Hawaii where the introduction of *Plasmodium relictum* played a role in the extinction of native birds [6, 7]. Even though these parasites are prevalent worldwide and cause extinction-level threats to naive birds, not much is known about the viruses that potentially infect these parasites.

Protists consist of a diverse umbrella of unicellular eukaryotic lineages including amoebas, phytoplankton's, *Leishmania*, and *Plasmodium* parasites [8, 9]. The most recent common ancestor between protists and multicellular eukaryotes was likely part of the origin of eukaryotes [10]. This creates an interest in studying the RNA viruses of protists as it may provide

**Data Availability Statement:** Raw sequence reads for 24 avian blood samples used in this experiment can be found on the NCBI SRA database under accession PRJNA529266 [9], and control data can be found with SRA ID numbers SRR7554463,

SRR7554428, SRR7554429, SRR7554464 [4]. Raw sequence reads for three Eurasian siskins (Carduelis spinus) infected with P. ashfordi can be found under bio project accession PRJNA311546 [26]. Raw data on the four nestling buzzards searched in this study can be found under bioproject accession number PRJEB5722 [20]. Raw sequence reads for two Eurasian siskins infected with P. delichoni and P. homocircumflexum can be found under bioproject accession PRJNA380974 and PRJNA343386, respectively [28]. RdRp sequences are included in the supplementary files and can be found with Genbank Accession numbers BK059437 for MaRNAV3, and BK059438 for MaRNAV4.

**Funding:** RS This work was supported by the grant to RNMS, NIH 1SC3GM118210-01A. Ravinder Sehgal grants.nih.gov JRR This work was supported by the Genentech Foundation Scholarship of the SEO office at San Francisco State University Jose Roberto Rodrigues The funders had no role in study design, data collection and analysis, decision to publish, or preparation of the manuscript.

**Competing interests:** The authors have declared that no competing interests exist.

information on the common ancestors of RNA viruses today [9]. Since it is believed that protists constitute all major eukaryotic lineages (supergroups) that diverged from the last common eukaryotic ancestor, it is possible that the direct predecessors of modern protists were present when the ancestors of multicellular eukaryotic viruses emerged [9]. Furthermore, many protists have remained in aquatic environments which suggests that they have evolved accompanied by their viromes. Aquatic environments can allow for virus survival through protection from UV radiation and desiccation. Taking into account all these evolutionary factors, it can be implied that protists are host to some of the most ancient RNA viruses that infect eukaryotes [9, 10].

In 2020, metatranscriptomic studies of human malaria parasites revealed the presence of an RNA virus that is suspected to infect the human malaria parasite, *P. vivax*. This RNA virus was named Matryoshka RNA virus 1 (MaRNAV1) and was only found in blood metatranscriptomes that were also infected with *Plasmodium vivax* [11]. The Russian term, Matryoshka, or "Russian doll" refers to how the RNA virus infects a parasite that in turn infects a vertebrate host. To assign the host of MaRNAV1, Charon et al. (2020) followed four assumptions: (i) high viral loads should only occur when the host parasite is also found in high amounts, (ii) the host parasite must be present when the virus is found, (iii) the virus has to be phylogenetically related to other previously identified viruses that infect similar parasite taxa, (iv) codon usage analysis should show similar results between the virus and its suspected host. In the same study, researchers used the RNA-dependent RNA-polymerase (the most conserved protein found in all RNA viruses) of MaRNAV1 to screen transcriptomes of avian samples infected with *Leucocytozoon* parasites, leading to the discovery of a second novel RNA virus named MaRNAV2 [11]. When the sequences for both of these viruses were aligned against the entire non-redundant protein database, the closest relatives were found to be RdRps of the Narnaviridae family. Viruses from this family are among the simplest biological entities and will often consist of only an RdRp as part of their genome [12]. Previously, narnaviruses have been discovered in arthropods, mosquitoes, fungi, and other protists [13]. The Wilkie narna-like virus 1 had the RdRp with highest amino acid similarity to MaRNAV1 and V2. The Wilkie narnalke virus 1 was isolated from mosquito metatranscriptomes, but researchers suspected that it likely infected a parasite in the mosquito and not the mosquito itself [14, 15].

Little is known about the effects of viruses that infect intracellular eukaryotic parasites. In another system, researchers examined the possible effects that Leishmania RNA virus 1 (LRV1) could have on *Leishmania* parasites. They found that the presence of the RNA virus enhances the pathogenicity of leishmaniasis in the vertebrate host [16]. These results show the importance of studying the interactions between RNA viruses and their single-celled hosts. Identifying these viruses in avian blood parasites could provide an ideal study system since birds and these parasites are found in natural ecosystems worldwide. Our objective here was to use available avian blood RNA data from the NCBI SRA database to search for the presence of RNA viruses associated with haemosporidian parasites. We identified two novel RNA viruses, one in *Leucocytozoon* infected birds and one in an individual infected with *Haemoproteus (Parahaemoproteus)*. These findings expand the number of the Matryoshka viruses to include all three of the major haemosporidian host genera.

## Methods

Using bioinformatic tools (Trinity, Trimmomatic, Diamond BLASTx), we performed local sequence alignments on the metatranscriptomes against the RdRps of RNA viruses of interest, including MaRNV1 and MaRNV2, which are suspected to infect *Plasmodium* and *Leucocytozoon* parasites, respectively. The raw SRA datasets used in this study were from blood samples

of 24 birds captured and sampled across three locations in North America including: Alaska, New Mexico, and New York [17]. All 24 birds were previously confirmed positive for *Plasmodium*, *Leucocytozoon*, and/or *Haemoproteus* (*Parahaemoproteus)* parasites by metatranscriptomic analysis [18]. We also searched for RNA viruses in nine avian blood samples available on the NCBI SRA database that consisted of avian blood samples infected with *P. ashfordi*, *P. delichoni*, *P. homocircumflexum*, and *Leucocytozoon* parasites [19–21].

## Transcriptome assembly

Raw sequence reads of 33 avian blood samples were downloaded from NCBI SRA by using fastq-dump to obtain in fastq format. Transcriptome assembly and quality trimming of the raw RNA reads were done by using Trinity Software v2.10.0 with Trimmomatic v0.40, using default parameters.

## Virus discovery

A local sequence-based homology search was implemented using Diamond v0.9.24 BLASTx against protein databases that contained RNA-dependent RNA polymerases (RdRp) of interest. All RdRp protein sequences used in this study were downloaded from the NCBI protein database using accession codes. Each assembled transcriptome was searched using Diamond BLASTx against a database containing the RdRp sequences and other segments from the newly discovered Matryoshka RNA virus 1 (MaRNAV1) and Matryoshka RNA virus 2 (MaRNAV2) [11]. These same transcriptomes were then searched using Diamond BLASTx against all available RdRp sequences from NCBI in order to screen for the presence of any similar RNA viruses that may have been present. For each candidate hit identified, we performed a NCBI BLASTx search against the entire non-redundant protein database to find similar sequences. To further decrease the likelihood of the suspected RdRp proteins from being EVEs (viral elements integrated into the host genome), we performed an NCBI BLASTn search and found no evidence of them being EVEs. As controls, we used four *P.vivax* transcriptomes from Charon et al. (2020). Of these four, three were infected with MaRNAV1 and one was free of infection from either the parasite or the virus. All three positive control transcriptomes resulted in hits of MaRNAV1 at ~98% identity, while the negative control transcriptome resulted in no hits of MaRNAV1.

Amino acid sequences were determined using NCBI ORF finder which translated nucleotide sequences in all six frames to determine open reading frames (ORF). NCBI ORF finder was used under parameters: minimum ORF length of 75 nucleotides (nt), genetic code standard, and using the ORF start codon of "ATG" only. The longest ORFs for MaRNAV3 and 4 were used for future analysis. Amino acid sequences of MaRNAV3 and 4 were submitted to the Protein Homology/Analogy Recognition Engine v 2.0 (Phyre2) web portal for structural-based homology searches and 3D structure prediction [22]. Both MaRNAV3 and 4 were submitted into the InterPro database for functional analysis by family classification and domain prediction [23].

## Phylogenetic analysis

To further analyze the suspected RNA viruses discovered, we produced a phylogenetic tree encompassing all the virus RdRp sequences that were included in the study conducted by Charon et al. (2020) and added our candidate RdRp sequences. First, we retrieved sequences of RdRps from the NCBI database using the NCBI accession codes included in the phylogenetic tree that was produced by Charon et al. (2020). Downloaded sequences were then aligned using MAFFT v7.309 with the E-INS-I algorithm. The aligned sequences were then

reformatted into Phylip|Phylip4 format so that they could be input into IQ-TREE for correct model selection [24]. The phylogenetic tree was outputted in Newick format. To view the phylogenetic tree, we used ggtree (R) and other R packages (ggplot2, treeio).

## Results

We studied 33 avian blood metatranscriptomes infected with *Plasmodium*, *Leucocytozoon*, and *Haemoproteus* (*Parahaemoproteus*). From these metatranscriptomes we found three with the presence of novel Matryoshka RNA viruses (Table 1). The first, found in a bird infected with *Leucocytozoon*, we named MaRNAV3, which had 59.5% amino acid identity to MaRNAV1 and 63.0% amino acid identity to MaRNAV2. The second virus, which we named MaRNAV4, had 44.7% amino acid identity to MaRNAV1 and 47.0% amino acid identity to MaRNAV2 (Table 1). To our knowledge, MaRNAV4 is the first known virus to be associated with the genus *Haemoproteus*. This also makes MaRNAV3 & V4 only the third and fourth viruses discovered to be associated with the parasites of the Apicomplexa subclass haemosporidia.

MaRNAV3 was discovered in a sample from an individual Alaskan songbird, Common redpoll (*Acanthis flammea*), that was infected with *Leucocytozoon* parasites [18]. The longest ORF identified was 994 AA in length and had 59.5% and 63.0% identity to MaRNAV1 and V2, respectively. When the longest ORF was input into InterPro for protein classification, it resulted in a match for DNA/RNA polymerase superfamily (IPR043502), which includes RdRps. When the longest ORF was run through Phyre2 in intensive mode for structure and function prediction, we had results for DNA/RNA polymerase superfamily with 27% modeled at >90% confidence (Table 1). In the same bird sample, we found positive hits for a second ORF that was 298 AA in length. The function of this ORF is unknown; however, it did have significant similarity to the second segments of MaRNAV1 and V2. This was another indication that MaRNAV3 is related to MaRNAV1 and V2 (S1 Table in S1 File).

MaRNAV4 was found in two birds of the same species found in New Mexico (*Plumbeous vireo* (*Vireo plumbeus*)) that were infected with *Parahaemoproteus* parasites. The longest ORF associated with this virus is 977 AA in length and was 37–44%, identical to the RdRp sequences of MaRNAV1 and V2 (Table 1). For this suspected RdRp, we acquired no hits to any family domain when submitted into the InterPro database for analysis. The AA sequence was put through Phyre2 in intensive mode for structure and function prediction; we had results for DNA/RNA polymerase with 18% modeled at >90% confidence (Table 1). The longest ORF

**Table 1. Data on specific hits of interest found in the 33 avian blood samples.** SRR8792714 & SRR8792718 hits had > 98% similarity, so we decided to use the longest ORF to represent as a single viral RdRp.

| Hits to Viral RdRps from NCBI SRA dataset | | | |
|---|---|---|---|
| **Name/ SRA number** | **MaRNAV4 SRR8792714** | **MaRNAV4 SRR8792718** | **MaRNAV3 SRR8792722** |
| **Host Bird Species** | *Vireo plumbeus* | *Vireo plumbeus* | *Acanthis flammea* |
| **Parasite(s) Present** | *Parahaemoproteus* | *Parahaemoproteus* | *Leucocytozoon* |
| **Length of nt Sequence** | 3083 nt | 2864 nt | 3257 nt |
| **Percent Identity to Matryoshka RdRps (%)** | 37–44% | 42–47% | 42–63% |
| **Diamond BLASTx hits to RdRp** | Hits for MaRNAV1, MaRNAV2, and Wilkie-narna-like Virus 1 RdRps | Hits for MaRNAV1, MaRNAV2, and Wilkie-narna-like Virus 1 RdRps | Hits for MaRNAV1(59.5%), MaRNAV2 (63%), and Wilkie-narna-like Virus 1 (42.2%) RdRps |
| **Longest ORF** | 977 aa | 938 aa | 994 aa |
| **InterPro Results (short names)** | None | None | DNA/RNA_pol_sf, |
| **Phyre2 Results** | 18% modeled with >90% confidence for DNA/RNA polymerases | 18% modeled with >90% confidence for DNA/RNA polymerases | 27% modeled with >90% confidence for DNA/RNA polymerase |

was also put into NCBI BLASTp against the entire non-redundant protein database. The proteins with the highest similarity to MaRNAV4 were narnaviruses that were associated with protists or arthropods (Fig 1), which strongly suggest that this was in fact a virus infecting *Parahaemoproteus* parasites, and not the vertebrate host. For MaRNAV4 we did not find the presence of any ORFs related to the second segments of MaRNAV1 and V2.

To further investigate the newly discovered viral sequences, we performed phylogenetic analysis that included the two novel RNA viruses described here. In addition, we included MaRNAV1, MaRNAV2 and all of their other closest relatives identified through BLASTp against the entire non-redundant NCBI protein database. Phylogenetic analysis confirmed

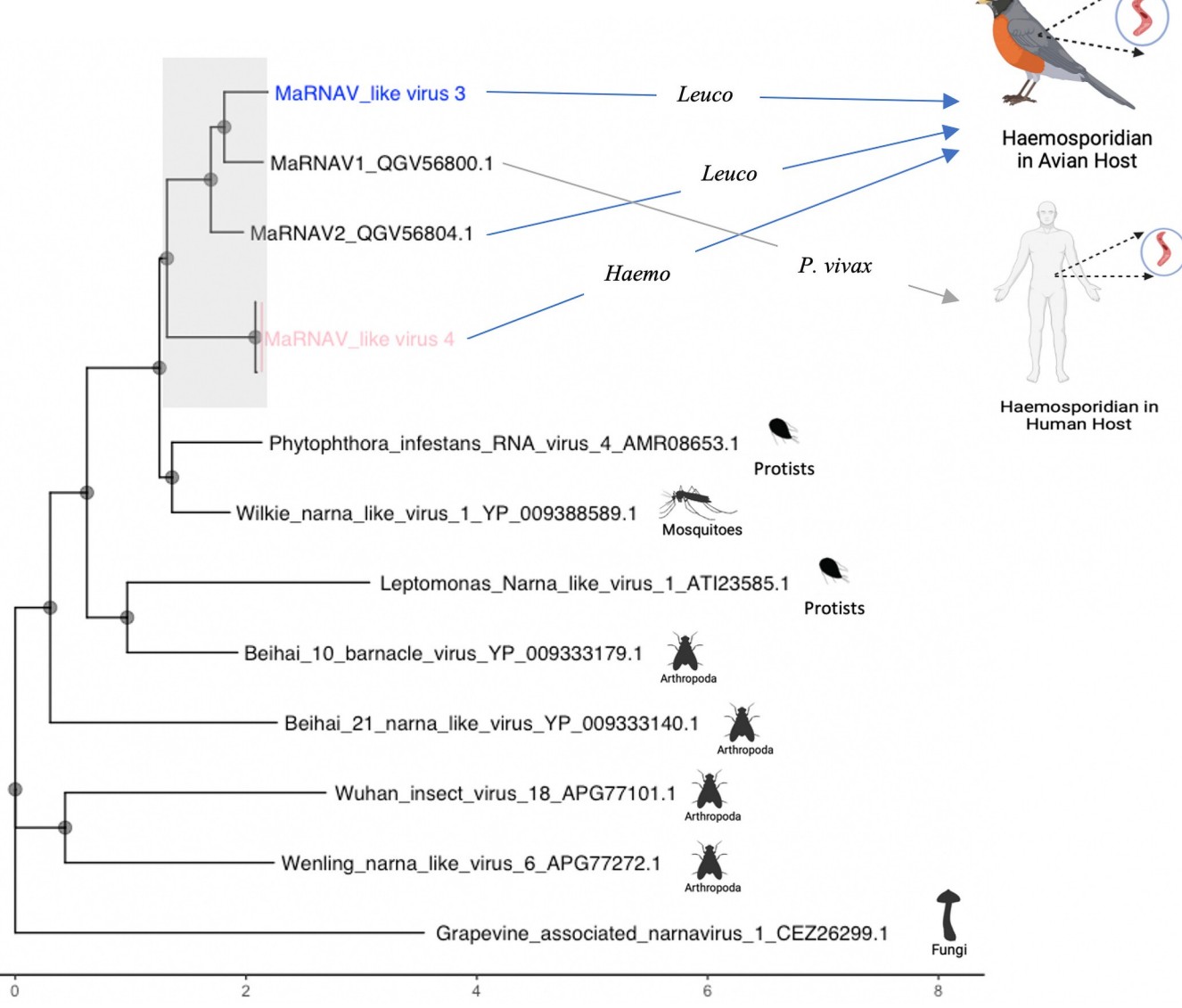

**Fig 1. Phylogeny of possible RNA viruses associated with avian haemosporidian parasites.** Phylogenetic tree comparing the most closely related RdRps of MaRNAV3 and MaRNAV4 from the NCBI non-redundant protein database. The images on the right side are the expected host or environments where RNA viruses were discovered (Icons from BioRender.com). Sequences were aligned using the E-INS-I algorithm in MAFFT v7.309. Aligned sequences were put into IQ-tree v1.6.10, which chose the LG+F+I+G4 model according to BIC. The scale refers to the number of nucleotide substitutions per codon site, and the tree is unrooted although the taxon Grapevine associated narnavirus is drawn at the root. Created with BioRender.com.

that MaRNAV3 and MaRNAV4 were closely related to the other Matryoshka RNA viruses, and that they belong in their own clade, separate from other narna-like viruses (Fig 1).

## Discussion

We have identified two new members of the Matryoshka viruses that appear to infect the avian haemosporidian genera *Haemoproteus* and *Leucocytozoon*. The segment found for both MaR-NAV3 and MaRNAV4 was a single ORF that consisted of a conserved RdRp-motif related to the motifs found in *Narnaviridae*, including MaRNAV1 and MaRNAV2. The *Narnaviridae* family consists of capsid-less viruses that have been suspected to infect plants, fungi, and protists [13]. We suspect that MaRNAV3 and MaRNAV4 have avian haemosporidian hosts because of their close relationship to other viruses associated with haemosporidians (Fig 1). MaRNAV3 is most closely related to MaRNAV1 and MaRNAV2 (Fig 1), which is expected as it is associated with *Leucocytozoon* parasites, similar to MaRNAV2. MaRNAV4 is less related to MaRNAV1 and MaRNAV2 (Fig 1), which is also to be expected, as the suspected host is *Parahaemoproteus*. The two main reasons we believe that MaRNAV3 and MaRNAV4 infect haemosporidian parasites are: (i) MaRNAV3 and V4 fall in the same clade as both MaRNAV1 and V2, which have been only found in the presence of their respective haemosporidian hosts, (ii) NCBI Blastx of MaRNAV3 and V4 results in the top hits coming from RdRp sequences from narna-like viruses that were found in arthropod or protist environments. However, further experimental studies would be necessary to confirm that the hosts of these viruses are haemosporidian parasites, and not the bird or other organisms inside the bird. Our results suggest a relatively low prevalence of these viruses, as we were only successful in finding three samples with positive hits from 33 metatranscriptomes, 24 from North America, and nine from Europe and Russia for MaRNAV3 and MaRNAV4. However, they have been found at higher prevalence in samples from Australia (8/12 samples) where these viruses where identified using both transcriptome analysis and rt-PCR for confirmation [11].

Since RNA viruses are known to have high rates of mutations, it is unlikely that the viruses described diverged through co-speciation as we would expect the amino acid identity to be much lower than ~41–60% [17]. In the same vein, avian haemosporidia phylogenetic analysis has shown that avian *Plasmodium* parasites are more closely related to *Haemoproteus*, than *Leucocytozoon* parasites [25]. Their respective associated viruses instead show that *Plasmodium* and *Leucocytozoon* parasites are more closely related than *Haemoproteus* parasites (Fig 1). Our data support the hypothesis by Charon et al. (2019) that Matryoshka RNA viruses evolve through viral cross-species transmission events because their parasite host will use arthropods as a vector, which can harbor a large number of parasites and bacteria. The vectors of these parasites likely are essential to these viral cross-species transmission events, as insects will often interact with a large number of parasites and vertebrates, which allows for the possible transfer of genetic material during co-infection [11, 26]. The high mutation rates would also help explain why Matryoshka RNA viruses are so divergent even when they infect the same genus, as in the case of MaRNAV2 and MaRNAV3 where both are associated with *Leucocytozoon* parasites but only have 63.0% amino acid identity. The discovery of these Matryoshka viruses associated with parasites of the genera *Leucocytozoon* and *Haemoproteus* in North America supports the theory that they are widespread since previously MaRNAV2 had only been found in bird samples from Australia [11]. More data would be necessary to discern the time-scale and occurrence of viral cross-species events [11].

Even though *Narnaviridae* are relatively simple viruses containing a single segment encoding for their RdRp, they may be able to impact their hosts in complex ways. There are three possible effects of an RNA virus on parasite-host relationships. The first is hypervirulence,

where a virus can increase the pathogenicity of the tandem virus-parasite on the host [27]. The second is hypovirulence, which refers to a decrease in the pathogenicity of the parasite on its host [27]. The third is that the viruses will use the parasite as a vector to enter the vertebrate host [27]. Noted examples of hypervirulence have been shown in the cases of the genera *Leishmania*, *Trichomonas*, and *Cryptosporidium*, where the presence of a virus increases the pathogenic effects of the parasite [27]. Previous studies have also identified instances where the removal of the virus has caused hypovirulent effects on the parasite, such as in *Giardia* where the presence of *G. lamblia* virus (GLV) caused growth arrest of the parasites [27]. Free-living amoebas will often come into contact with humans and thus be rendered parasitic, and since protists can harbor viruses, they can be viewed as vectors into human hosts. This has been seen in the case of adenoviruses (causative agents of diarrhea, pneumonia, and conjunctivitis), where free-living amoebas protected the virus from harsh environments and acted as carriers into the hosts [27]. We currently do not know how Matryoshka RNA viruses affect the parasite-host relationship of haemosporidian parasites. More experimental work should be done to define these relationships. Since we have only found evidence of these viruses using bioinformatics methods, it is difficult to make any assumptions about MaRNA viruses. If MaRNA viruses cause hypovirulence, they could be useful in the creation of anti-malaria therapeutics. Further investigation is necessary to understand the relationship of these possibly significant viruses for the treatment of haemosporidian infections.

It will be important to continue to search for these Matryoshka-like viruses in birds so that we can further understand their prevalence and diversity. It is still unclear if these RNA viruses are associated with all haemosporidians in all birds or only a select few: presently they have only been found in birds infected with *Leucocytozoon* or *Haemoproteus*. We presume that avian *Plasmodium* species should also be infected. However, although the virus was found in human samples infected with *P. vivax*, they were not found associated with *P. falciparum* [11]. Electron microscopy has revealed virus-like particles in the avian parasite genera *Leucocytozoon*, *Parahaemoproteus* and *Plasmodium* [28]. It would be interesting to ascertain whether these crystalloid inclusions were actually examples of Matryoshka viruses. Future studies to learn more about the infection biology of these RNA viruses would require isolation of an infected parasite and inoculating a healthy bird to analyze the gene expression levels of the birds throughout infection. This would give us clues into what molecular mechanisms these MaRNA viruses use to infect their parasite hosts. At this point, we have found evidence of these RNA viruses in three haemosporidian families, which suggests that they are widespread in a variety of different parasites in various locations around the globe.

## Supporting information

**S1 Fig. InterPro results for longest ORF of RdRp like sequence associated with *Leucocytozoon*.** Shows Homology to the DNA/RNA polymerase super family.
(TIF)

**S2 Fig. Percent identity matrix of MaRNA viruses.** We found that two samples had similar RdRp sequences (~98% amino acid identity), hence we described them both as MaRNAV4.
(TIF)

**S1 File. S1 and S2 Tables controls data and information on protein of unknown function associated with MaRNAV3.** Results from datasets used to test pipeline used in this study. Results of protein with unknown function including InterPro submission data.
(PDF)

**S1 Appendix. Diamond BLASTx results, IQ-Tree files, MaRNAV3 V4 RdRp sequences, Phyre2 structure prediction pdb files, MaRNAV3 second segment data, and Matryoshka viruses alignment.** Results from diamond BLASTx using two databases provided, and an E-value cutoff of 1E-10. Includes Trinity assembly stats report for all transcriptomes used in this study. IQ-Tree files include Newick format tree file and aligned sequences used for analysis. (ZIP)

## Acknowledgments

We are grateful to Leilani Nguyen, Rachel Quock, Faith De Amaral for support with reviewing the manuscript.

## Author Contributions

**Conceptualization:** Jose Roberto Rodrigues, Scott W. Roy, Ravinder N. M. Sehgal.

**Data curation:** Jose Roberto Rodrigues.

**Formal analysis:** Jose Roberto Rodrigues, Ravinder N. M. Sehgal.

**Funding acquisition:** Ravinder N. M. Sehgal.

**Investigation:** Ravinder N. M. Sehgal.

**Methodology:** Jose Roberto Rodrigues, Scott W. Roy.

**Supervision:** Scott W. Roy, Ravinder N. M. Sehgal.

**Visualization:** Jose Roberto Rodrigues, Scott W. Roy.

**Writing – original draft:** Jose Roberto Rodrigues, Scott W. Roy, Ravinder N. M. Sehgal.

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
