## [Decision Letter · Decision Letter 0]

12 Jan 2022

PONE-D-21-37085Novel RNA viruses associated with avian haemosporidian parasitesPLOS ONE

Dear Dr. Rodrigues,

Thank you for submitting your manuscript to PLOS ONE. After careful consideration, we feel that it has merit but does not fully meet PLOS ONE’s publication criteria as it currently stands. Therefore, we invite you to submit a revised version of the manuscript that addresses all the points raised during the review process.

We look forward to receiving your revised manuscript.

Kind regards,

Maria Andreína Pacheco, Ph.D.

Academic Editor

PLOS ONE

Journal Requirements:

(RS

This work was supported by the grant to RNMS, NIH 1SC3GM118210-01A. 

Ravinder Sehgal

grants.nih.gov

JRR

This work was supported by the Genentech Foundation Scholarship of the SEO office at San Francisco State University 

Jose Roberto Rodrigues

The funders had no role in study design, data collection and analysis, decision to publish, or preparation of the manuscript.)

Please include your amended Funding Statement within your cover letter. We will change the online submission form on your behalf."

4. We note that Figure 1 in your submission contain copyrighted images. All PLOS content is published under the Creative Commons Attribution License (CC BY 4.0), which means that the manuscript, images, and Supporting Information files will be freely available online, and any third party is permitted to access, download, copy, distribute, and use these materials in any way, even commercially, with proper attribution. For more information, see our copyright guidelines: http://journals.plos.org/plosone/s/licenses-and-copyright.

Reviewers' comments:

Reviewer's Responses to Questions

**Comments to the Author**

1. Is the manuscript technically sound, and do the data support the conclusions?

Reviewer #1: Yes

Reviewer #2: Yes

2. Has the statistical analysis been performed appropriately and rigorously? 

Reviewer #1: N/A

Reviewer #2: N/A

3. Have the authors made all data underlying the findings in their manuscript fully available?

Reviewer #1: Yes

Reviewer #2: Yes

4. Is the manuscript presented in an intelligible fashion and written in standard English?

Reviewer #1: Yes

Reviewer #2: No

5. Review Comments to the Author

Reviewer #1: The overall design of the study is acceptable and can conduct to the conclusion that new MaRNAV-like signals are identified as narna-like viruses and very likely infect bird hosts. Nevertheless, I would recommend to the authors to be more careful with the virus/host relationship establishment because, as well explained in the Discussion part, this study and the previous one from Charon et al are both based on bioinformatic identification. The use of "confirm" l. 197 is thus not adapted. On the other hand, some Narnaviridae are not only "suspected" fungal viruses as indicated l. 217 but have been clearly and experimentally associated to host infection (especially in S. cerevisiae).

Discussion of the identification of second segment is missing, as well as the discussion of the potential presence of the virus in the host genome. I would recommend to the authors to conduct a Blastn of their newly identified sequences to the nt NCBI to ensure the newly identified viruses are not EVEs, i.e. viral elements integrated into host genomes.

Please see below for minor comments/corrections :

l.25 : Precise the taxon of parasites here ? (Apicomplexa/Haemosporidian)

l.29 : Does it mean MaRNAV-3 share 40% with MaRNAV-1 and MaRNAV3 share 60% with MaRNAV-2 ? Unclear...

l.56 : I would add another ref for the protist definition and phylogeny... Perhaps Burki F. et al. (2020) ‘The New Tree of Eukaryotes’, Trends in Ecology & Evolution, 35: 43–55.

l.76 : RdRp cannot be considered as "highly" conserved but constitutes the most conserved protein in RNA viruses.

l.83 : Narnaviruses have been previously identified in protists (Oomycetes and Leishmania).

l.141: Indicate the parameters of ORF Finder (which genetic codes have been used for example). it can be critical when searching for protist viruses related to Lenarviricota as some could be mitochondria-associated.

l.161 : I would not use the the term "infection" but rather "presence", "association". As detailed in the Discussion part, bioinfo-based virus identification do not formally demonstrate host infection...

l.184 : What is the level of ID shared between the second segments of MaRNAV1 and 2 ? S2 table mention only one percentage for both MaRNAV1 and 2... No additional segments could be found for MaRNAV4 ?

l.197 : Replace "confirm" by "strongly suggest" ?

l.252 : The study in Charon et al has screened several SRA files from P. vivax Bioprojects to conclude that MaRNAV1 is widespread in P. vivax samples worldwide (not only in Malaysian samples).

Table1 : Percent of identity with what ?

Figure1: The quality is bad. Is the tree rooted ? What is the scale at the bottom ?

Supplemental material : I would suggest sharing the alignment in fasta format. Could you share the Phyre2 info tables as well ?

Reviewer #2: This is an interesting report. Perhaps the discussion is a little speculative, but I understand the need of providing a general context of otherwise a descriptive study. Some minor changes

The phrase: “Even though protists are no less evolved than more complex multicellular eukaryotes, they play an important role in deducing the origins of RNA viruses today”

I understand the intention, but even mentioning a “scala naturae” to dismiss it without arguing against do not assist the reader without background in evolutionary biology. This is not the place to discuss it, but it is not simply a stylistic matter so I found such a phrase a little unsettling. The issue could be addressed by simply stating that the most recent common ancestor between protists and multicellular eukaryotes likely was part of the origin of eukaryotes and as such, study the RNA viruses in the extant lineages of protist and multicellular eukaryotes may inform about the common ancestor of the RNA viruses observed today.

Lines between 255-275 are too speculative, I suggest changing it by emphasizing, briefly, the need of experimental work to establish the viruses effect, if any, in their hosts (in this case the Haemosporida parasites).

6. PLOS authors have the option to publish the peer review history of their article (what does this mean?). If published, this will include your full peer review and any attached files.

Reviewer #1: No

Reviewer #2: No

---

## [Decision Letter · Decision Letter 1]

30 May 2022

Novel RNA viruses associated with avian haemosporidian parasites

PONE-D-21-37085R1

Dear Dr. Rodrigues,

We’re pleased to inform you that your manuscript has been judged scientifically suitable for publication and will be formally accepted for publication once it meets all outstanding technical requirements.

Kind regards,

Maria Andreína Pacheco, Ph.D.

Academic Editor

PLOS ONE

Reviewers' comments:

Reviewer's Responses to Questions

**Comments to the Author**

1. If the authors have adequately addressed your comments raised in a previous round of review and you feel that this manuscript is now acceptable for publication, you may indicate that here to bypass the “Comments to the Author” section, enter your conflict of interest statement in the “Confidential to Editor” section, and submit your "Accept" recommendation.

Reviewer #2: All comments have been addressed

2. Is the manuscript technically sound, and do the data support the conclusions?

Reviewer #2: Yes

3. Has the statistical analysis been performed appropriately and rigorously? 

Reviewer #2: Yes

4. Have the authors made all data underlying the findings in their manuscript fully available?

Reviewer #2: Yes

5. Is the manuscript presented in an intelligible fashion and written in standard English?

Reviewer #2: Yes

6. Review Comments to the Author

Reviewer #2: There are still a typos in the first page. I do not understand the use of the statement "These authors contributed equally to this work" and only appears one, which happen to be the corresponding. Perhaps it was intended for the other two authors.

7. PLOS authors have the option to publish the peer review history of their article (what does this mean?). If published, this will include your full peer review and any attached files.

Reviewer #2: No

---

## [Editor Report · Acceptance letter]

22 Jun 2022

PONE-D-21-37085R1 

Novel RNA viruses associated with avian haemosporidian parasites. 

Dear Dr. Rodrigues:

I'm pleased to inform you that your manuscript has been deemed suitable for publication in PLOS ONE. Congratulations! Your manuscript is now with our production department. 

Kind regards, 

on behalf of

Dr. Maria Andreína Pacheco 

Academic Editor

PLOS ONE